# High Frequency of Undiagnosed Psychiatric Disorders in Inflammatory Bowel Diseases

**DOI:** 10.3390/jcm9051387

**Published:** 2020-05-08

**Authors:** Irene Marafini, Lucia Longo, Damun Miri Lavasani, Rodolfo Rossi, Silvia Salvatori, Federica Pianigiani, Emma Calabrese, Alberto Siracusano, Giorgio Di Lorenzo, Giovanni Monteleone

**Affiliations:** 1Gastroenterology Unit, Department of Systems Medicine, University of Rome Tor Vergata, 00133 Rome, Italy; irene.marafini@gmail.com (I.M.); silviasalvatori23@gmail.com (S.S.); pianigianifederica@gmail.com (F.P.); emma.calabrese@uniroma2.it (E.C.); 2Psychiatriy Unit, Department of Systems Medicine, University of Rome Tor Vergata, 00133 Rome, Italy; lucialongo2@yahoo.it (L.L.); damunmiri@yahoo.it (D.M.L.); rudy86.rossi@gmail.com (R.R.); siracusano@med.uniroma2.it (A.S.); 3IRCCS Santa Lucia Foundation, 00133 Rome, Italy

**Keywords:** Crohn’s disease, ulcerative colitis, mood disorders, PTSD

## Abstract

Inflammatory bowel diseases (IBD) are associated with comorbid depressive and anxiety disorders, but a significant proportion of IBD patients with psychiatric disorders (PsychD) remain undiagnosed and untreated. The aim of this study was to assess the frequency and type of undiagnosed PsychD in IBD patients. Two hundred and thirty-seven adult IBD (136 Crohn’s disease (CD) and 101 ulcerative colitis (UC)) outpatients were consecutively recruited at a single university hospital centre between January 2018 and June 2019. After a scheduled follow-up visit for IBD clinical evaluation, participants underwent a semi-structured interview with a trained psychiatrist. One hundred and fourteen (48%) IBD patients had at least one PsychD, and in 67 (59%) of them, a diagnosis was made for the first time during the study. The remaining 47 (41%) patients had received a previous psychiatric diagnosis, but in only six cases was it consistent with the diagnosis made during the study. PsychD were equally distributed in CD (72/136, 53%) and UC (42/101, 42%), and mostly represented by mood disorders (54/114, 47%) and anxiety (27/114, 24%) disorders. PsychD were not related to the disease severity, phenotype or localisation of IBD, even though having three or more concomitant PsychD was associated with more severe disease in CD. Our data indicate that undiagnosed PsychD are common in IBD and highlight the importance of considering psychiatric evaluation in the management of IBD patients.

## 1. Introduction

Inflammatory bowel diseases (IBD) are chronic and disabling disorders of the gastrointestinal tract that include Crohn’s disease (CD) and ulcerative colitis (UC) [1,2,3]. The aetiologies of both CD and UC are still unknown, but many environmental factors are supposed to act in genetically-predisposed individuals and trigger an immune-inflammatory process leading to IBD-associated tissue damage [4]. It has been also shown that some environmental factors, such as stress and life events, can adversely affect mucosal homeostasis, thereby influencing IBD natural history [5,6].

When compared with the general population, patients with CD and patients with UC are 2–4 times more likely to develop depression in their lifetime and 3–5 times more likely to develop anxiety disorders [7,8,9,10,11,12]. There is also evidence that such psychiatric disorders (PsychD) are independent predictors of quality of life in both teenage and adult patients with IBD regardless of the severity of the condition, and consistently, the use of anti-depressants and cognitive behavioural therapies has a beneficial effect on the quality of life of IBD patients [13,14]. Nonetheless, prior studies suggest that depression and anxiety disorders remain underdiagnosed in most IBD patients [15,16], and an accurate estimate of the real frequency of PsychD in IBD is lacking as most data were derived from databases and self-report questionnaires, without proper clinical evaluations. It is also unclear whether additional PsychD other than depression and anxiety are frequent in IBD.

Untreated mental illnesses could have important implications for IBD patients beyond their impact on quality of life, as some studies documented an association between PsychD and the course of IBD [17]. Indeed, IBD patients with PsychD have more symptoms, an emotional representation of illness, higher perceived stress, less activity engagement, and a greater need for gastroenterologist visits [18]. These associations highlight the importance of the appropriate and timely recognition and treatment of PsychD in the IBD population.

The aim of this study was to evaluate the presence and types of undiagnosed PsychD in IBD patients using a semi-structured interview with a trained psychiatrist.

## 2. Methods

### 2.1. Ethical Considerations

Each patient that took part in the study gave written informed consent, and the study protocol was approved by the local Ethics Committees (Tor Vergata University Hospital, Rome).

### 2.2. Participants

Consecutive outpatients with a confirmed diagnosis of IBD were prospectively recruited between January 2018 and May 2019 at a single university hospital (University of Rome Tor Vergata). The participants were aged 18 years or older, able to provide informed consent and able to complete a structured clinical interview.

### 2.3. Study Procedures

During the gastroenterological visit, information regarding demographic characteristics, medical and surgical history, disease severity and current therapy of IBD was collected. In particular, medical therapy taken at the time of the clinical and psychiatric evaluation was registered, while there was no documentation of previous psychiatric treatments. Disease characteristics were defined according to the Montreal classification. Disease severity was assessed using the Harvey–Bradshaw Index (HBI) for CD and partial Mayo Score (PMS) for UC. Remission was defined as an HBI <5 or a Mayo score <2 [19,20]. Patients with segmented colitis associated with diverticulosis were excluded.

After the visit with the gastroenterologists, participants completed a psychopathological assessment with a psychiatrist trained in conducting the Italian version of Mini International Neuropsychiatric Interview 5.0.0 (MINI 5.0.0) [21]. The MINI 5.0.0 provides psychiatric diagnoses based on the Diagnostic and Statistical Manual of Mental Disorders IV Text revision (DSM-IV-TR) criteria through a structured interview that in a general practice setting lasts for 15 min, with a very high inter-rater reliability [22].

### 2.4. Statistical Analysis

Demographic and clinical variables of interest were compared between CD and UC using χ^2^ tests or *t*-tests as appropriate. Comparisons between the demographic and clinical characteristics of CD patients and UC patients with a PsychD diagnosis and those without any PsychD was done using χ^2^ tests or *t*-tests. Psychiatric diagnoses were grouped into broad diagnostic categories as follows: unipolar depression (major depressive disorder, dysthymia, and mood disorder with psychotic manifestations), bipolar disorders (bipolar I and bipolar II), anxiety disorders (generalised anxiety disorder, social phobia, panic disorder and agoraphobia), post-traumatic stress disorder (PTSD), abuse and dependence disorders (alcohol, cocaine and cannabinoids) and obsessive-compulsive disorder (OCD). The relative prevalence of PsychD in CD and UC patients was estimated. The independence of the broad psychiatric diagnostic categories and UC and CD diagnoses was assessed using χ^2^ tests with the relative contribution estimate of each cell. In the next step, we grouped patients according to the number of concomitant PsychD they endorsed (no diagnosis and 1, 2 and 3 or more diagnoses) and tested the independence of this grouping variable from CD/UC membership. The relationship between CD and UC disease severity and the number of PsychD was assessed using an ordered logistic regression model, separately for CD and UC, and the data were expressed as odds ratios (ORs) and 95% Confidence Intervals (95% CIs). All statistical analysis was performed with STATA^®^ 16. *p*-Values ≤ 0.05 were considered statistically significant.

## 3. Results

### 3.1. High Frequency of Undiagnosed Psychiatric Disorders in IBD

Overall, 258 consecutive participants with IBD were recruited, of whom 237 (91.8%) agreed to participate in the study. The demographic and clinical data of the IBD patients are shown in Table 1. Of the 237 patients enrolled, 101 (43%) were UC and 136 (57%) were CD. The two groups did not statistically differ according to gender proportion, age or smoking habits. One hundred and fourteen (48.16%) patients received a diagnosis of PsychD; 72/114 (63.16%) had CD and 42/114 (36.84%) had UC, with no significant differences between the two diseases.

Further analysis revealed that in 67/114 (50 CD and 17 UC; 59%) patients, the PsychD was diagnosed for the first time during the study. The remaining 47 patients (22 CD and 25 UC, 41%) had received a diagnosis of PsychD in the past, but in only six of them was the diagnosis was consistent with that made during the study.

Next, we compared the demographic and clinical characteristics of CD patients (Table 2) and UC patients (Table 3) with a PsychD diagnosis with those of patients without any PsychD. No differences between the two groups were seen in both CD and UC, except for a larger proportion of steroid users in CD with PsychD as compared to CD patients without PsychD, and for UC women with PsychD as compared to those without PsychD.

### 3.2. Analysis of Psychiatric Disorders in IBD

When analysis was restricted to the patients with PsychD, we found that 23 (20.17%) CD patients and eight (7%) UC patients received a diagnosis of unipolar depression, 13 (11.40%) CD patients and 10 (8.7%) UC patients were diagnosed with bipolar disorder, 15 CD patients (13.15%) and 12 UC patients (10.52%) received a diagnosis of anxiety disorders, and 16 (14.03%) CD patients and seven (6.14%) UC patients received a diagnosis of post-traumatic stress disorder (PTSD). A diagnosis of an abuse/dependence disorder was made in five (4.38%) CD patients and two (1.75%) UC patients while an obsessive-compulsive disorder was diagnosed in three (2.63%) UC patients.

Finally, we compared the frequency of each PsychD between CD and UC. Except for unipolar depression, which was more prevalent in CD than in UC, no differences were found (Table 4).

Twenty-five (58.14%) CD patients and 18 (41.86%) UC patients had a single PsychD, 29 (72.5%) CD patients and 11 (27.5%) UC patients had two concomitant PsychD, while 18 (58.06%) CD patients and 13 (41.94%) UC patients had three or more (up to six) concomitant PsychD. No diagnosis of anorexia was made, while five patients (four CD) met the criteria for bulimia.

With the ordinal logistic model, we showed that the number of concomitant PsychD was not associated with clinical severity in the UC patient group, while having three or more concomitant PsychD was associated with more severe disease in CD (Table 5).

## 4. Discussion

In the last few years, evidence has accumulated to suggest that patients with CD and patients with UC have increased risks of developing psychiatric comorbidities, which in turn are supposed to impair the quality of life of the patients and precipitate IBD flares [23]. Nonetheless, psychiatric evaluation is not routinely considered in the management of IBD patients. This study was undertaken to assess the frequency and type of PsychD in IBD. More than 90% of the consecutive IBD outpatients seen during a scheduled follow-up visit agreed to participate in the study, and nearly 50% of the participants had at least one PsychD. Notably, the majority of the PsychD patients were unaware of their mental illness, and in less than 15% of the patients who had a previous positive psychiatric evaluation was there consistency between the initial diagnosis and that made during the study. These data reinforce the concept that PsychD are common in IBD and remain, in many cases, undiagnosed.

Moreover, our findings confirm and expand on previous data showing that mood disorders and anxiety disorders are the most prevalent PsychD in IBD, since unipolar depression, bipolar disorders and anxiety disorders were diagnosed in more than two thirds of our patients, with no significant difference between CD and UC. We found that the frequency of anxiety and depression was significantly higher in women with UC than in their male counterparts. A similar trend was seen also in CD, but the data were not statistically significant. Overall, the data are consistent with the broader medical literature showing that anxiety and mood disorders are more common among women than men. PTSD was diagnosed in one fifth of the PsychD patients, in line with a recent study examining temporal trends in IBD incidence and prevalence in a US national cohort of veterans and showing a progressive increase in the prevalence of cases of anxiety, depression and/or PTSD during the study period [24].

Unlike many previous studies, which focused only on depressive disorders and anxiety disorders and in which psychiatric diagnosis was made through the use of questionnaires, our data are based on a semi-structured interview, which allowed us to diagnose psychiatric illness. Interestingly, nearly 50% of the PsychD patients had three or more concomitant disorders, which further underlines the necessity of an accurate psychiatric evaluation to better characterise mental illness.

The biological link between PsychD and IBD remains to be ascertained. Previous studies have shown that the onset of PsychD can precede the development of IBD [7], but it is unknown whether this reflects shared risk factors, which in genetically-predisposed individuals, trigger a final common inflammatory pathway leading to both PsychD and IBD. On the other hand, PsychD can arise in patients with established IBD, and circumstantial evidence suggests that immune-inflammatory factors released from the damaged guts of IBD patients or gut microbiota-derived neuroactive molecules could promote/exacerbate mental illness [25].

Our study indicates that many PsychD remain undiagnosed in IBD. Since the treatment of underlying PsychD has been associated with improvements in disease severity and quality of life in IBD, our findings support the beneficial effects of a proper psychiatric assessment and treatment in these patients [13,14]. However, further studies are needed to better address these issues.

The main strength of this study is that a psychiatrist performed the psychopathological assessment, while the majority of the previous studies in IBD patients were based upon self-report questionnaires. We are well aware that the study has some limitations as well. Most of the participants were in clinical remission and therefore could not reflect the general IBD population. Moreover, IBD severity was evaluated exclusively using clinical scores as laboratory parameters and/or endoscopic/histologic data were not available for all the patients at the time of the study. This leaves open the possibility that PsychD are more frequent in patients with active intestinal inflammation than in patients who are in stable biological remission as suggested by previous studies [26,27]. Moreover, our study lacks a control group, which makes difficult to establish the exact risk of any PsychD for patients with IBD.

## 5. Conclusions

In conclusion, our data indicate that in IBD, there is a high frequency of comorbid PsychD, which in many cases, remain undiagnosed. These findings highlight the importance of considering psychiatric evaluation in the management of IBD patients.

## Figures and Tables

**Table 1 jcm-09-01387-t001:** Demographic and clinical characteristics of the 237 enrolled patients.

Diagnosis	CD *n* (%)	UC *n* (%)
***n***	136 (57)	101 (43)
**Female gender**	70 (51)	39 (39)
**Mean age (years)**	42.98	41.79
**Smokers**	36 (26)	15 (15)
**CD onset age**		
A1	11 (8)	-
A2	85 (63)	-
A3	40 (29)	-
**CD location**		
L1	63 (46.3)	-
L2	16 (11.8)	-
L3	57 (41.9)	-
**CD behaviour**		
B1	60 (44)	-
B2	56 (41)	-
B3	20 (15)	-
**Perianal disease**	29 (21)	-
**Colitis extent**		
E1	-	11 (11)
E2	-	42 (42)
E3	-	47 (47)
**Disease severity**	HBI	PMS
Remission	110 (80.8)	62 (61.4)
Mild disease	18 (13.2)	32 (31.7)
Moderate disease	7 (5)	6 (5.9)
Severe disease	1 (0.7)	1 (1)
**Positive surgical history**	65 (48)	6 (6)
**Therapy**		
Anti-TNF	22 (16)	7 (7)
Vedolizumab	3 (2)	1 (1)
Thiopurines	27 (20)	5 (5)
Steroids	22 (16)	16 (16)
Mesalamine	76 (56)	80 (79)

CD: Crohn’s disease; UC: ulcerative colitis; HBI: Harvey–Bradshaw Index; PMS: partial Mayo Score; TNF: tumor necrosis factor; -:non applicable.

**Table 2 jcm-09-01387-t002:** Demographic and clinical characteristics of Crohn’s disease (CD) patients with and without any current psychiatric disorders (PsychD).

Crohn’s Disease	PsychD *n* (%)	NO PsychD *n* (%)	*p*-Value
***n***	72	64	
**Female gender**	30 (41.6)	19 (29)	0.312
**Mean age (years)**	42	45	0.239
**Smokers**	21 (29)	15 (23)	0.450
**CD onset age**			
A1	9 (12)	2 (3)	0.961
A2	45 (63)	40 (63)	
A3	18 (25)	22 (34)	
**CD location**			
L1	33 (46)	30 (47)	0.961
L2	10 (14)	7 (11)	
L3	29 (40)	27 (42)	
**CD behaviour**			
B1	33 (46)	27 (42)	0.848
B2	28 (39)	28 (44)	
B3	11 (15)	9 (14)	
**Perianal disease**	19 (26)	10 (16)	0.126
**Disease severity**			
Remission	56 (78)	54 (84)	0.224
Mild disease	10 (14)	8 (12)	
Moderate disease	6 (8)	1 (2)	
Severe disease	-	1 (2)	
**Positive surgical history**	34 (47)	31 (48)	0.887
**Therapy**			
Biologic therapy	15 (21)	10 (16)	0.560
Steroids	16 (22)	6 (9)	0.05

**Table 3 jcm-09-01387-t003:** Demographic and clinical characteristics of ulcerative colitis (UC) patients with and without any current psychiatric disorders (PsychD).

Ulcerative Colitis	PsychD *n* (%)	NO PsychD *n* (%)	*p*-Value
***n***	42	59	
**Female gender**	21 (50)	18 (31)	0.047
**Mean age (years)**	41	42	0.779
**Smokers**	21 (50)	7 (12)	0.309
**UC extent**			
E1	6 (17)	5 (9)	0.347
E2	14 (33)	28 (47)	
E3	20 (50)	26 (44)	
**Disease severity**			
Remission	26 (62)	36 (61)	0.662
Mild disease	13 (31)	19 (32)	
Moderate disease	2 (5)	4 (7)	
Severe disease	1 (2)	-	
**Positive surgical history**	2 (5)	5 (8)	0.469
**Therapy**			
Biologic therapy	4 (10)	4 (7)	0.615
Steroids	5 (12)	11 (19)	0.36

**Table 4 jcm-09-01387-t004:** Psychiatric diagnoses grouped into broad diagnostic categories for Crohn’s disease (CD) and ulcerative colitis (UC).

IBD Patients *n* = 237	CD *n* (%) 136	UC *n* (%) 101	*p*-Value
Current psychiatric disorders (PsychD)			
No PsychD	64 (52.03)	59 (47.97)	0.083
Unipolar depression	23 (74.19)	8 (25.81)	0.042
Bipolar disorders	13 (56.52)	10 (43.47)	0.92
Anxiety disorders	15 (55.56)	12 (44.44)	0.83
Post-traumatic stress disorder	16 (69.57)	7 (30.43)	0.21
Abuse/dependence disorders	5 (71.43)	2 (28.57)	0.44

**Table 5 jcm-09-01387-t005:** Relationship between Crohn’s disease (CD) and ulcerative colitis (UC) disease severity and the number of concomitant current psychiatric disorders (PsychD).

Concomitant PsychD	CD Disease Severity HBI	UC Disease Severity PMS
	OR (95% CI)	OR (95% CI)
1	1.03 (0.29, 3.63)	0.46 (0.13, 1.58)
2	1.16 (0.35, 3.7)	0.81 (0.22, 3.01)
≥3	3.5 (1.13, 11.03)	2.49 (0.78, 7.93)

OR: odd ratio; CI: confidence intervals.

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
