# Peer review of "High Frequency of Undiagnosed Psychiatric Disorders in Inflammatory Bowel Diseases"

_jcm, 2020, doi:10.3390/jcm9051387_

Round 1
Reviewer 1 Report
Congratulations to the authors of this manuscript.
The appealing aspect of this study is that the diagnosis of a psychological disorder was made by a psychiatrist using a structured interview. This is different to many other studies which typically use questionnaires or chart review to establish a diagnosis of a psychiatric disorder. Further the interesting finding is that there were many patients with an undiagnosed psychological disorder.
There are some aspects to this manuscript I would suggest that the authors change/ consider reassessing.
- Page 3, line 106 (subtitle results). The authors say 258 patients were "recruited" and then followed by 237 "agreed" to participate.
- Does this mean there were patients who were not "recruited" which were not considered in this cohort and offered the study? Does this IBD clinic only have 258 patients?
- Page 3, line 110 (subtitle results). The authors present the data as
"patients received diagnosis of PD: 72/114 (30.37%) had CD and 42/114 (17.79%) had UC"
- They should reword this as it is automatically confusing when 72/114 = 63%. Shouldn't the denominator me the number of CD and UC patients respectivley?
- Page 5, "Analysis of psychiatric disorder": Did the authors examine for eating disorders, avoidant food restrictive disorder with their structured interview, there is data suggesting this is common among patients with GI disorders broadly including FGID's and IBD.
- Page 6 Discussion, Line 173 - 190. Could the authors consider rewording, shortening this paragraph? Or consider removing it entirely?
- This study does not evaluate anything to do with possible links between PD and the development of IBD, yet a large chunk in the discussion has been allocated to this.
- Could the authors consider expanding on relevance of undiagnosed IBD, relationship between Tx of psych disorders and improvement in QoL in IBD, relevance of QoL in IBD in relation to comorbid PD, how comorbid PD in IBD influences healthcare use in IBD - all of these would be relevant given you have proven a hidden world of undiagnosed PD in IBD.
- Page 6 Discussion
- Could the authors expand on the benefits/trade offs of using a psychiatrist to consecutivley diagnose PD - i would argue this is the strength of this study.
Well done on this study
Author Response
Referee 1:
We would like to thank the reviewer for his/her positive evaluation and helpful suggestions. In response to the specific issues raised by this reviewer:
Congratulations to the authors of this manuscript. The appealing aspect of this study is that the diagnosis of a psychological disorder was made by a psychiatrist using a structured interview. This is different to many other studies, which typically use questionnaires or chart review to establish a diagnosis of a psychiatric disorder. Further, the interesting finding is that there were many patients with an undiagnosed psychological disorder. There are some aspects to this manuscript I would suggest that the authors change/consider reassessing.
- Page 3, line 106 (subtitle results). The authors say 258 patients were "recruited" and then followed by 237 "agreed" to participate. Does this mean there were patients who were not "recruited" which were not considered in this cohort and offered the study? Does this IBD clinic only have 258 patients?
Response: Of the 258 IBD patients consecutively enrolled only 237 gave consent to undergo the psychiatric interview. The IBD center follows more than 258 patients, but the psychiatric evaluation was restricted to patients routinely followed by the authors (gastroenterologists) of the manuscript and additional patients managed by other clinicians of the team were not enrolled.
Page 3, line 110 (subtitle results). The authors present the data as "patients received diagnosis of PD: 72/114 (30.37%) had CD and 42/114 (17.79%) had UC". They should reword this as it is automatically confusing when 72/114 = 63%. Shouldn't the denominator me the number of CD and UC patients respectively?
Response: Line 110: we revised the text taking into account the reviewer’s comment.
- Page 5, "Analysis of psychiatric disorder": Did the authors examine for eating disorders, avoidant food restrictive disorder with their structured interview, there is data suggesting this is common among patients with GI disorders broadly including FGID's and IBD.
Response: During the psychiatric interview, eating disorders were also assessed. No diagnosis of anorexia was made, while 5 patients met the criteria for bulimia. This information has been included in the revised manuscript.
- Page 6 Discussion, Line 173 - 190. Could the authors consider rewording, shortening this paragraph? Or consider removing it entirely? This study does not evaluate anything to do with possible links between PD and the development of IBD, yet a large chunk in the discussion has been allocated to this.
Could the authors consider expanding on relevance of undiagnosed IBD, relationship between Tx of psych disorders and improvement in QoL in IBD, relevance of QoL in IBD in relation to comorbid PD, how comorbid PD in IBD influences healthcare use in IBD - all of these would be relevant given you have proven a hidden world of undiagnosed PD in IBD. Page 6 Discussion. Could the authors expand on the benefits/trade offs of using a psychiatrist to consecutivley diagnose PD - i would argue this is the strength of this study. Well done on this study.
Response: We have substantially revised the discussion section. As suggested by the reviewer, the parts dedicated to the possible links between psychiatric disorders and the development of IBD has been shortened and we have expanded on the relevance of diagnosing psychiatric disorders in IBD patients.

Reviewer 2 Report
- This is an interesting study and its strength lies in the use of psychiatric semi-structured interview for quite a large number of patients. The findings are also interesting as they suggest higher psychiatric morbidity than previously thought which may be due to the more detailed assessment in this study, than other studies that have used self-report questionnaires. The authors should add that to the discussion. I think it is also worth commenting that most of the patients' disease was quiescent and although they found some association between PD and disease activity in CD, this wasn't very strong.
- Overall it is well written.
- The authors should state whether they carried out a power calculation.
- I am a little concerned about the numbers and percentages...some of them seem wrong...so the authors need to check these
- Line 110 check % 72/114 is not 30.37 % and 42/114 is not 17.79%
- Check Table 4 the % again seem wrong
- The abbreviation PD is usually used to refer to personality disorder, so the authors may wish to use a different term PsychD for instance.
Author Response
Referee 2:
We would like to thank the reviewer for his/her positive evaluation and helpful suggestions. In response to the specific issues raised by this reviewer:
- This is an interesting study and its strength lies in the use of psychiatric semi-structured interview for quite a large number of patients. The findings are also interesting as they suggest higher psychiatric morbidity than previously thought which may be due to the more detailed assessment in this study, than other studies that have used self-report questionnaires. The authors should add that to the discussion. I think it is also worth commenting that most of the patients' disease was quiescent and although they found some association between PD and disease activity in CD, this wasn't very strong.
Response: As suggested by the reviewer, we have expanded the discussion paragraph highlighting the strengths of the study. We have also discussed the fact the most patients had an inactive disease at the time of the psychiatric evaluation.
- Overall it is well written.
- The authors should state whether they carried out a power calculation.
Response: A power calculation of the study was not made. The intent of the study was to provide a punctual description of the prevalence of psychiatric disorders in IBD.
I am a little concerned about the numbers and percentages...some of them seem wrong...so the authors need to check these.
Line 110 check % 72/114 is not 30.37 % and 42/114 is not 17.79%
Check Table 4 the % again seem wrong
Response: all the percentages were re-calculated and we made the suggested changes.
- The abbreviation PD is usually used to refer to personality disorder, so the authors may wish to use a different term PsychD for instance.
Response: We have changed the abbreviation PD with PsychD.
